# Tracing neuronal circuits in transgenic animals by transneuronal control of transcription (*TRACT*)

Ting-hao Huang[1], Peter Niesman[1], Deepshika Arasu[1], Donghyung Lee[1], Aubrie L De La Cruz[1], Antuca Callejas[1,2], Elizabeth J Hong[1], Carlos Lois[1]*

[1]Division of Biology and Biological Engineering, California Institute of Technology, Pasadena, United States; [2]Department of Cell Biology, School of Science, University of Extremadura, Badajoz, Spain

**Abstract** Understanding the computations that take place in brain circuits requires identifying how neurons in those circuits are connected to one another. We describe a technique called TRACT (*TRA*nsneuronal *C*ontrol of *T*ranscription) based on ligand-induced intramembrane proteolysis to reveal monosynaptic connections arising from genetically labeled neurons of interest. In this strategy, neurons expressing an artificial ligand ('donor' neurons) bind to and activate a genetically-engineered artificial receptor on their synaptic partners ('receiver' neurons). Upon ligand-receptor binding at synapses the receptor is cleaved in its transmembrane domain and releases a protein fragment that activates transcription in the synaptic partners. Using TRACT in *Drosophila* we have confirmed the connectivity between olfactory receptor neurons and their postsynaptic targets, and have discovered potential new connections between neurons in the circadian circuit. Our results demonstrate that the TRACT method can be used to investigate the connectivity of neuronal circuits in the brain.

DOI: https://doi.org/10.7554/eLife.32027.001

*For correspondence:
clois@caltech.edu

**Competing interests:** The authors declare that no competing interests exist.

## Introduction

Comprehensively mapping the connectivity of diverse neural circuits across many brain regions and many organisms is a major goal of modern neuroscience (*Denk et al., 2012*; *Bargmann and Marder, 2013*; *Swanson and Lichtman, 2016*). In addition, recent research indicates that aberrant neuronal wiring may be the cause of several neurodevelopmental disorders (*Peça and Feng, 2012*; *Rubinov and Bullmore, 2013*; *Mevel and Fransson, 2016*), further emphasizing the importance of identifying the wiring diagrams of brain circuits. To address this issue, several new approaches have been developed. Each of these methods has strengths and limitations, which are discussed in detail in a recent review (*Lee et al., 2017*).

In a previous work, we demonstrated that a strategy based on ligand-induced intramembrane proteolysis can be used to monitor cell-cell interactions in the nervous system of transgenic *Drosophila* (*Huang et al., 2016*). In this strategy, cells expressing an artificial ligand ('donor' cells) bind to and activate an artificial receptor on their interacting partners ('receiver' cells) via interactions across the intercellular space. We previously used this method to investigate the interactions between neurons and glial cells in the *Drosophila* central nervous system (*Huang et al., 2016*). However, in its original implementation, the system was not useful to trace neuronal circuits because it revealed all forms of cell-cell contact, including non-synaptic contacts.

To allow this strategy to trace neuronal circuits it is necessary to ensure that the interaction of ligand and receptor occurs exclusively across synapses. To prevent activation of the system between neurons that had non-synaptic cell-cell contacts, we engineered the ligand so that it would be

**eLife digest** One of the main obstacles to understanding how the brain works is that we know relatively little about how its nerve cells or neurons are connected to one another. These connections make up the brain's wiring diagram. Current methods for revealing this wiring all have limitations. The most popular method – serial electron microscopy – can reveal the connections in a small region of the brain in great detail, but it cannot show connections between neurons that are far apart.

Huang et al. have now created a genetic system for visualizing these connections. For neurons to communicate, one neuron must produce a signal called a ligand. This ligand can then bind to and activate its partner neuron. Huang et al. modified the DNA of neurons so that every time those cells produced a specific ligand, they also produced a red fluorescent protein. Similar modifications ensured that every time the ligand activated a partner neuron, the activated neuron produced a green fluorescent protein. Viewing the red and green neurons under a microscope enabled Huang et al. to see which cells were communicating with which others.

While these experiments took place in fruit flies, the same approach should also work in other laboratory animals, including fish, mice and rats. Once we know the wiring diagram of the brain, the next step is to investigate the role of the various connections. To understand how a computer works, for example, we might change the connections between its circuit components and look at how this affects the computer's output. With this new method, we can change how neurons communicate with one another in the brain, and then look at the effects on behavior. This should provide insights into the workings of the human brain, and clues to what goes wrong in disorders like schizophrenia and autism.

DOI: https://doi.org/10.7554/eLife.32027.002

selectively located at the presynaptic cleft (*Südhof, 2012*; *Van Vactor and Sigrist, 2017*). To validate that the optimized ligands revealed neurons specifically connected by synapses, we tested the system on the *Drosophila* antennal lobe, an olfactory center with well-understood connectivity established by light and electron microscopy, and electrophysiological methods (*Grabe et al., 2016*; *Stocker et al., 1990*; *Wilson, 2013*; *Rybak et al., 2016*). We observed that adding domains from the synaptic proteins synaptobrevin (nSyb) or syndecan (sdc) was sufficient to achieve reliable synaptic tracing in the antennal lobe. In addition, we used TRACT to investigate the connectivity of the *Drosophila* circadian system, a compact brain circuit with a small number of neurons (*Dubowy and Sehgal, 2017*; *Nitabach and Taghert, 2008*), and discovered new candidate postsynaptic targets for the PDF neurons of the circadian system in the central brain, some of which that express the circadian-related gene, *per*. These results demonstrate that the TRACT method will be a useful addition to the arsenal of methods available to investigate the connectivity of brain circuits.

## Results

In our previously published work, we described an artificial ligand-receptor system based on intramembrane proteolysis to investigate connections between cells (*Huang et al., 2016*). In short, upon ligand-receptor interaction at sites of cell-cell contact the transmembrane domain of an engineered receptor is cleaved by intramembrane proteolysis and releases a protein fragment that regulates transcription in the interacting partners (*Figure 1a*, *Gordon et al., 2015*; *Morsut et al., 2016*; *Huang et al., 2016*). In its original implementation, we used a ligand called CD19mch that contains the extracellular and transmembrane domains (ECD and TMD) of the mouse lymphocyte antigen CD19 (*Fujimoto et al., 1998*) fused to the red fluorescent protein mCherry in its intracellular domain (ICD). The original artificial receptor is called SNTG4 and contains the following domains (in N-to-C terminal order): (i) the ECD from a single chain antibody (ID3) that recognizes mouse CD19 (*Kochenderfer et al., 2009*), (ii) the Notch regulatory region (NRR) and the TMD from *Drosophila* Notch (spanning from EGF repeat 36 until the ICD) (*Kovall et al., 2017*), and (iii) esn, a simplified version of the yeast transcriptional activator Gal4 (*Figure 1a*) (*Sprinzak et al., 2010*). In this configuration, expression of the ligand and receptor in neurons leads to the distribution of these two proteins uniformly throughout the membrane, including cell body, dendrite branches, and axons. To

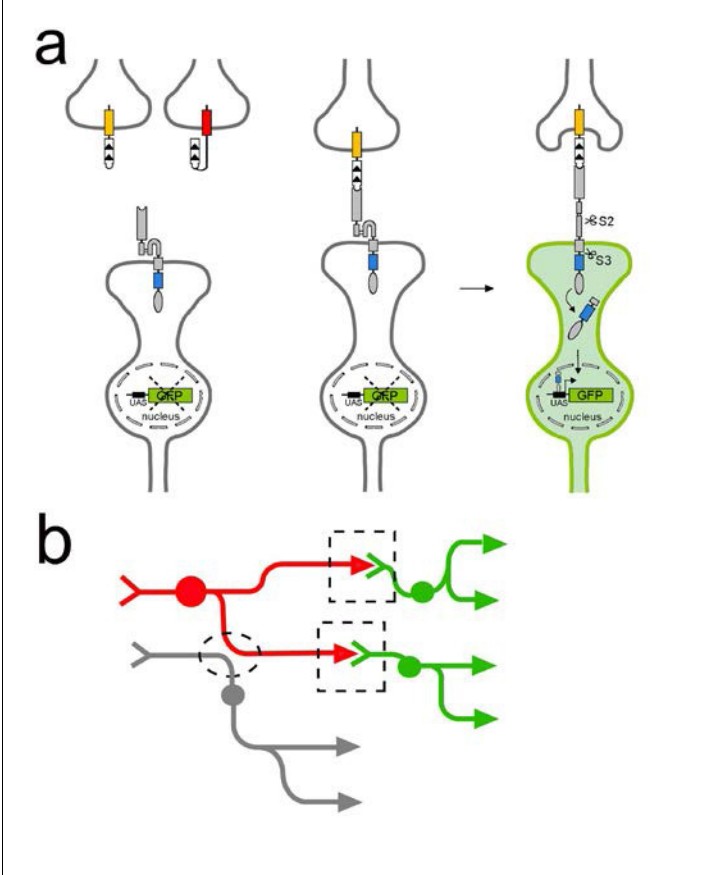

**Figure 1.** TRACT: Using ligand-induced intramembrane proteolysis to reveal circuits of neurons connected by synapses. (a) Ligand-induced intramembrane proteolysis to monitor synaptic contacts between neurons. Sequence of events upon ligand-receptor interaction are depicted from left to right. The ligand domain, (CD19, white domain with the arrowheads indicating the orientation of protein sequence from N- to C-terminal) is localized to the presynaptic plasma membrane in the 'donor' neuron of interest (top) by fusing it to domains from two different presynaptic proteins, Sdc (yellow rectangle) or nSyb (red rectangle). The receptor (grey and blue rectangles, and grey oval) is targeted to synaptic sites by introducing the intracellular domain (blue rectangle) of neuroligin (NLGN) between the transmembrane domain of Notch and the transcription factor Gal4 (gray oval). When the ligand binds to the receptor on a synaptic partner neuron (bottom), it partially unfolds the notch regulatory region (NRR) to allow for cleavage of the receptor in the S2 site by endogenous metalloproteases. S2 cleavage shortens the receptor, and induces a second cleavage (S3) by intramembrane proteolysis (mediated by ɣ-secretase) that liberates the transcription factor (gray oval) that is part of the intracellular domain of the receptor. This transcription factor then translocates to the nucleus to activate transcription of reporter genes such as GFP. (b) Using TRACT to detect synaptic connections between neurons. For anterograde tracing, the ligand is localized on the presynaptic neurons of interest ('donor' neuron labeled in red). The receptor is expressed in all potential synaptic partners ('receiver' neurons labeled in green and gray). GFP expression will be activated (green) only in the receiver neurons that make synaptic contacts (indicated by stippled boxes) with the donor neurons (red). In contrast, GFP expression should not be activated in neurons (gray) that are in close proximity to donor neurons but do not make synapses (stippled oval).

DOI: https://doi.org/10.7554/eLife.32027.003

detect the activation of the receptor upon interaction with its ligand in vivo in *Drosophila*, we included a UAS-GFP reporter transgene, which expresses GFP in response to Gal4 activity (*Brand and Perrimon, 1993*).

To test whether the expression of the ligand in donor neurons could reveal the subset of receiver neurons that receive synaptic input from them, we focused on the antennal lobe, the second-order olfactory processing area in the *Drosophila* brain that receives direct input from the primary olfactory receptor neurons (ORNs) (*Figure 2a*, *Wilson, 2013*; *Laissue and Vosshall, 2008*). Synapses in the

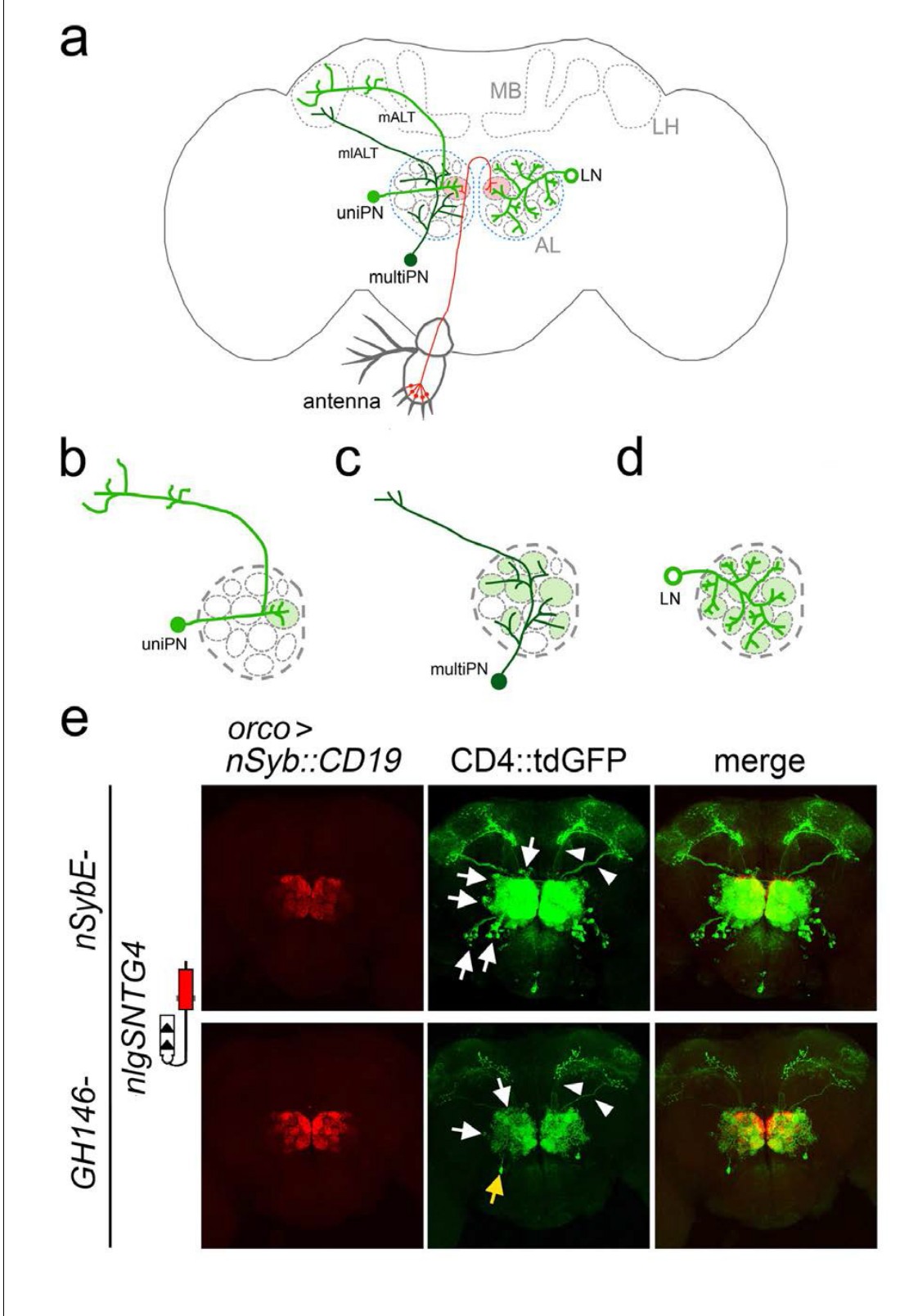

**Figure 2.** Connections between olfactory receptor neurons and antennal lobe neurons revealed by TRACT. (a) Olfactory receptor neurons (ORNs) labeled in red, have their cell bodies located into two peripheral sensory organs, antennae (illustrated here) and maxillary palps, and have axons that project into the antennal lobe (AL) in the brain. All axons from ORNs expressing the same olfactory receptor converge in two glomeruli (one in each hemisphere) within the AL (red circles). There are two main types of neurons (green) in the antennal lobe, projection neurons (PNs) and local neurons

*Figure 2 continued on next page*

*Figure 2 continued*

(LNs). PNs have dendrites that branch in glomeruli and axons that project towards the mushroom body (MB) and/or the lateral horn (LH). (b, c) There are two types of PNs, uniglomerular PNs (uniPNs) and multiglomerular PNs (multiPNs). (b) The dendrites of uniPNs branch in a single glomerulus, and their axons project into the MB and LH via the medial antennal lobe tract (mALT). (c) The dendrites of multiPNS branch into multiple glomeruli, and their axons project into the LH via the mediolateral ALT (mlALT). (d) LNs are axonless neurons and their dendrites branch into many (or most) of the glomeruli. (e) Detection of synaptic contacts between olfactory receptor neurons and antennal lobe neurons in the adult *Drosophila* antennal lobe with the nlgSNTG4 receptor driven by *nSybE* enhancer (top panels) and GH146 enhancer (bottom panels). Induction of GFP expression in neurons (arrows) surrounding the antennal lobe when the ligand (nSyb::CD19) was driven by the *orco* driver in ORNs. Arrowheads indicate the axons of PNs in mALT and mlALT. Left: nSyb::CD19+ axons from ORNs (red); middle: GFP+ neurons in the antennal lobe (green); right: merged images of nSyb::CD19 and GFP. In the brains with the nlgSNTG4 receptor driven by the nSybE enhancer (top panels) GFP expression was induced in different neuronal types, including uniPNs, LNs and and several multiPNs at the ventral part. In the brain with the nlgSNTG4 receptor driven by the PN-specific driver GH146 (bottom panels), only PNs were GFP+. Most of these cells are uniPNs, but there was also one GFP+ multiPN in each antennal lobe (yellow arrow). Maximum projection of z-stack confocal images. Scale bar = 50 µm. See *Supplementary file 2* for additional information.

DOI: https://doi.org/10.7554/eLife.32027.004

The following figure supplements are available for figure 2:

**Figure supplement 1.** TRACT reveals antennal lobe neurons that have cell-cell contacts with olfactory receptor neurons.

DOI: https://doi.org/10.7554/eLife.32027.005

**Figure supplement 2.** Expression of the nlgSNTG4 receptor under the SybE enhancer.

DOI: https://doi.org/10.7554/eLife.32027.006

**Figure supplement 3.** Expression of CD19 fused with different presynaptic proteins in ORNs.

DOI: https://doi.org/10.7554/eLife.32027.007

**Figure supplement 4.** Comparison of localization of the CD19::Nrx, nSyb::CD19 and CD19::sdc ligands into presynaptic sites in the ORNs targeting DA1.

DOI: https://doi.org/10.7554/eLife.32027.008

**Figure supplement 5.** Identity of the postsynaptic targets of olfactory receptor neurons in the antennal lobe detected by TRACT.

DOI: https://doi.org/10.7554/eLife.32027.009

**Figure supplement 6.** TRACT with non-synaptically localized ligand reveals neurons that are not exclusively connected by synapses.

DOI: https://doi.org/10.7554/eLife.32027.010

**Figure supplement 7.** Control brains without the lexA driver to assess the levels of ligand-independent background.

DOI: https://doi.org/10.7554/eLife.32027.011

antennal lobe are organized into discrete compartments, called glomeruli, and each glomerulus corresponds to a distinct ORN class (defined by the odorant receptor it expresses) (*Berck et al., 2016*). All ORNs of a specific class project their axons to a common glomerulus (*Vosshall et al., 1999*; *Fishilevich and Vosshall, 2005*; *Couto et al., 2005*). The principal second-order olfactory neuron is called a projection neuron (PN). The vast majority of PNs are uniglomerular PNs (uniPNs) (*Figure 2b*), and each of these uniPNs have dendrites that branch into single glomeruli where they receive direct synaptic input from a single class of ORNs. Axonal output from uniPNs projects to two major third-order olfactory areas, the mushroom body and the lateral horn, via the medial antennal lobe tract (mALT, formerly iACT) (*Ito et al., 2014*), (*Figure 2a*). In addition, a smaller subset of PNs, multiglomerular PNs (multiPNs) (*Figure 2c*) (*Stocker et al., 1990*; *Parnas et al., 2014*; *Liang et al., 2013*), have dendritic arbors which pool input across multiple glomeruli in the antennal lobe, and send their output to the lateral horn only via the mediolateral ALT (mlALT, formerly the mACT) (*Figure 2a*) (*Tanaka et al., 2012*). Finally, a large set of local neurons (LNs) send and receive synaptic input from both ORNs and PNs, densely interconnecting the glomeruli (*Figure 2d*) (*Chou et al., 2010*; *Hong and Wilson, 2015*; *Seki et al., 2010*).

Expression of the original TRACT ligand and receptor (without any domains that would localize them to synaptic sites) in neurons failed to reveal any synaptic connections in the antennal lobe. For instance, when the CD19mch ligand is expressed in the majority of ORNs using the *Orco-lexA* driver (*Stocker et al., 1997*) and the SNTG4 receptor is expressed from the *elav* enhancer (*Yao and White, 1994*), we predicted labeling of both PNs and LNs, the major postsynaptic targets in the antennal lobe. However, in flies carrying *Orco > CD19* mch, *elav-SNTG4, UAS-GFP* transgenes, no GFP induction was observed in the antennal lobe (data not shown), consistent with the findings from another publication (*He et al., 2017*), suggesting that the system needed adjustments to detect synaptic connections.

To generate a genetic system that could be used to reliably identify synaptically connected neurons we tested a number of different modifications with the ligand, receptor, the drivers used to direct their expression, and reporters (discussed in full detail in the supplementary methods). The following modifications proved to be effective to achieve specific transneuronal labeling:

- To achieve panneuronal expression of the SNTG4 receptor we used the enhancer from the synaptobrevin gene (nSybE), and confirmed that this driver directs expression of the receptor in the vast majority of neuronal types in the brain at detectable levels (*Figure 2*, *Figure 2—figure supplement 1*, *Figure 2—figure supplement 2*).
- To allow for the immunodetection of the receptor, we added the V5 tag to the ICD of SNTG4 (*Figure 2*, *Figure 2—figure supplement 1*, *Figure 2—figure supplement 2*).
- To allow for the immunodetection of the ligand we added the OLLAS tag to the ICD of mCD19 (*Figure 2*, *Figure 2—figure supplement 3*).
- To achieve selective localization of the ligand into presynaptic sites we added the ICDs of syndecan or synaptobrevin into CD19 (CD19sdc and nSybCD19, respectively), and confirmed that these ICDs targeted mCD19 into presynaptic sites that co-localized with or were adjacent to the presynaptic marker BRP ((*Figure 2*, *Figure 2—figure supplement 4*).
- To achieve selective localization of the receptor into postsynaptic sites we added the ICD of the *Drosophila* neuroligin gene into SNTG4 (nlgSNTG4) (*Figure 1a*), and confirmed that this ICD enriched the localization of the SNTG4 receptor in the neuropil regions, which contain neuronal synapses ((*Figure 2*, *Figure 2—figure supplement 1*, *Figure 2—figure supplement 2*).
- To enable visualization of the dendrites and axons of neurons we used the reporter UAS-CD4tdGFP, in which tdGFP is localized into the plasma membrane (*Han et al., 2011*).

To test the ability of TRACT to reveal the connectivity originating from ORNs, we generated flies carrying *orco*>nSybCD19 or *orco*>CD19 sdc, nSybE-nlgSNTG4, and UAS-CD4tdGFP. In these flies, we observed GFP expression in dozens of neurons in the antennal lobe. Roughly half the GFP+ neurons were PNs, which can be identified by their axons projecting into the mushroom body and lateral horn, consistent with the known connectivity between ORNs and PNs (*Figure 2e*, top panels). In addition, the cell bodies of these axon-bearing neurons were located in the anterodorsal, lateral, and ventral sectors of the antennal lobe, consistent with them being PNs. Finally, most of the putative PNs were immunopositive for CHAT, a known marker of PNs (*Figure 2*, *Figure 2—figure supplement 5*). Around half the GFP+ neurons in the antennal lobe did not have axons, were immunopositive for GABA, and their cell bodies were located dorsolateral and ventrolateral, all features consistent with them being LNs (*Figure 2e*, top panels and *Figure 2*, *Figure 2—figure supplement 5*) (*Okada et al., 2009*).

The experiments with the *orco* driver revealed that the TRACT system can be used to activate gene expression in neurons known to be connected by synapses, but it did not prove that the LNs and PNs that were highlighted by TRACT were strictly connected by synapses to the ORNs expressing ligand (*Figure 1b*). In principle, any type of cell-to-cell contact, including non-synaptic membrane-to-membrane contact, could be sufficient to activate the TRACT receptor in the receiver neurons. We took advantage of the anatomy of the *Drosophila* antennal lobe to investigate whether TRACT could be used to reveal neurons solely connected by synapses. In *Drosophila*, all ORNs expressing the same olfactory receptor project their axons into two bilaterally symmetric glomeruli in the antennal lobe (*Grabe et al., 2016*; *Tanaka et al., 2012*). In addition, the dendrites of individual uniPNs branch into single glomeruli where they make synapses with the axons of ORNs (*Figure 2a and b*). Due to this feature of the connectivity of uniPNs, it is possible to unambiguously confirm that a given uniPN is connected to a single type of ORN through synaptic contacts that will exclusively occur in an identified glomerulus. Thus, to test the synaptic specificity of TRACT, we focused on the connectivity between ORNs and uniPNs. To selectively express the receptor in PNs, we generated transgenic flies in which the nlgSNTG4 receptor is driven by the GH146 enhancer, which is expressed in the majority of antennal lobe PNs (*Stocker et al., 1997*) (*Figure 2e* bottom panels, 3 and 4). To investigate the connectivity between ORNs and PNs, we generated flies carrying *orco*>nSybCD19, GH146-nlgSNTG4, and UAS-CD4::tdGFP (*Han et al., 2011*). In these flies we observed dozens of PNs (identifiable by their axons projecting into the mushroom body and lateral horn) with GFP expression in the antennal lobe, consistent with the known connectivity between ORNs and PNs (*Figure 2e*, bottom panels).

To investigate the ability of TRACT to reveal neurons exclusively connected by synapses we crossed GH146-nlgSNTG4 flies with flies that expressed the nSyb::CD19 and CD19::sdc ligands in identified glomeruli. If TRACT exclusively revealed neurons connected by synaptic contacts, the uniPNs whose receptors were activated by interaction with the ligand would have GFP+ dendrites that would selectively branch in the glomeruli onto where the ORNs would converge their axons. We crossed the GH146-nlgSNTG4 flies with flies expressing the nSyb::CD19 and CD19::sdc ligand in glomeruli VC1 and DA1/VA6/VA1lm under the enhancers of R28H10 and R17H02 LexA drivers, respectively (http://www.virtualflybrain.org) (*Figure 3*). We observed that with the nSyb::CD19 and CD19::sdc ligands, the uniPNs labeled projected their dendrites selectively into the glomeruli where the ligand was expressed, consistent with previous work (*Figure 3*) (*Grabe et al., 2016*). We counted the number of PNs induced by the nSyb::CD19 or CD19::sdc ligands when expressed in identified glomeruli, and found that the observed numbers matched those from previously published work (*Grabe et al., 2016*) (*Table 1a and b*). For example, a study using photoactivatable GFP (paGFP) indicated that between 8 and 10 PNs receive input from the DA1 glomerulus (*Grabe et al., 2016*), and when we expressed the nSyb::CD19 ligand with the R17H02-LexA driver in the DA1 glomerulus, we observed ~11 labeled uniPNs (*Figure 3a* and *Table 1a*). Similarly, previous work indicated that between 1–3 and 1–2 PNs receive input from the VA6 and VC1 glomeruli, respectively, and with the nSyb::CD19 ligand, we observed that one uniPN expressed GFP in each of these glomeruli (*Figure 3a* and *Table 1a*) (*Grabe et al., 2016*). In the VA1lm glomerulus, we did not observe any induction in PNs with the nSyb::CD19, which may be due to the low level of the receptor expression in this glomerulus (*Figure 3a*). When using the CD19::sdc ligand in identified glomeruli, we observed one PN labeled in VA6 and VC1 and ~12 PNs labeled in DA1 (*Figure 3b* and *Table 1b*).

A previous work demonstrated that, in addition to the uniPNs, the GH146 enhancer drives expression of transgenes into one multiPN (*Marin et al., 2002*). With the CD19::sdc and nSyb::CD19, in addition to the uniPNs described above, one multiPN was labeled with both R17H02 and R28H10 drivers, which has been identified previously (*Figures 3b* and *4* and *Table 1a and b*) (*Marin et al., 2002*). The dendrites of this neuron almost cover the whole antennal lobe, and its axon projects first to the lateral horn and then anterodorsally toward the midline. (*Figure 3c and d*).

Although we did not observe any differences in the number or distribution of uniPNs when the ligands were CD19::Sdc or nSyb::CD19, we observed that the labeling of multiPNs had different requirements for sdc and nSyb. Whereas we could achieve labeling of multiPNs with a single copy of CD19::sdc (in heterozygote animals) (*Figures 3b* and *4b*), we could only detect multiPNs when the nSyb::CD19 ligand was present in two copies (homozygote animals) (*Figures 3a* and *4a*). This observation suggests first, that the amount of ligand may be a limiting factor for transneuronal labeling, and second, that in some cases, the CD19::sdc could be more effective at revealing some synaptic partners than nSyb::CD19. These results indicate that both the nSyb::CD19 and CD19::sdc ligands selectively reveal neurons connected by synaptic contacts between ORNs and PNs, match the available data from previously published works, and confirm that TRACT can be used to perform anterograde transneuronal tracing of brain circuits in the *Drosophila* brain.

To explore the ability of the TRACT method to trace neuronal circuits in other brain areas, we investigated the brain circuits involved in the control of *Drosophila* circadian behavior (*Figure 5a*) (*Konopka and Benzer, 1971*; *Hall, 1998*; *Rosbash et al., 2003*; *Young, 1998*; *Nitabach and Taghert, 2008*). There are two key advantages of the circadian circuit to test the usefulness of the TRACT system: (i) it consists of a relatively low number of neurons (around 150), and (ii) the function and anatomy of many of those neurons are known in some detail, although their connectivity is not fully understood (*Beckwith and Ceriani, 2015*; *Peschel and Helfrich-Förster, 2011*; *Shafer et al., 2006*).

As a test case, we explored the ability of TRACT to reveal the connectivity of the PDF neurons, a well-characterized set of cells in the *Drosophila* brain that are critical regulators of circadian rhythm (*Figures 5b* and *6* and *Figure 5*, *Figure 5—figure supplement 1*; *Renn et al., 1999*; *Lear et al., 2009*). There are two types of PDF neurons in the *Drosophila* brain: (i) s-LNv neurons, which project their axons into the dorsal regions of the central brain, and (ii) l-LNv neurons, which have large dendrites that branch in the optic lobe, and an axon that projects to the contralateral optic lobe (*Figure 5a*). In addition to *pdf*, the period (*per*) gene is another critical regulator of circadian function in *Drosophila* (*Konopka and Benzer, 1971*; *Bargiello and Young, 1984*; *Reddy et al., 1984*). Previous works using GRASP suggested that s-LNv neurons were connected to the DN1p

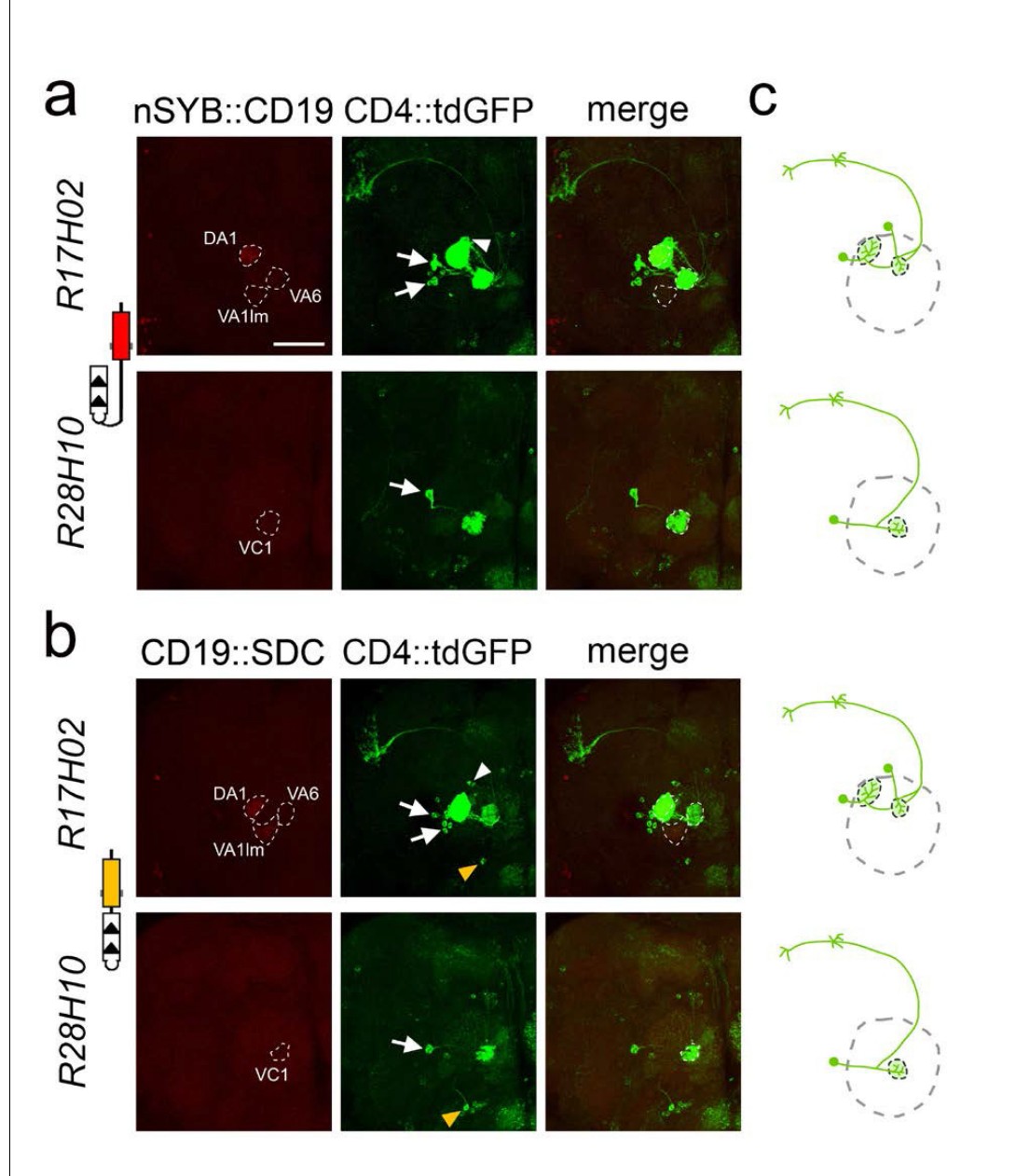

**Figure 3.** Selective labeling of PNs that receive synaptic input from ORNs in identified glomeruli. (a) Labeling of PNs that receive synaptic input from ORNs expressing the ligand nSyb::CD19 in identified glomeruli using the R17H02 (top panels) and R28H10 (bottom panels) LexA drivers. Left: nSyb:: CD19+ axons from ORNs (red) driven by R17H02- and R28H10-LexA branch in identified glomeruli (stippled circles). In R17H02 the nSyb::CD19 expression level was low in VA6 and VA1lm, and expression in DA1 was only visible after signal amplification by immunostaining. Center: Induction of CD4::tdGFP expression in PNs triggered by nSyb::CD19+ ORNs (red, left panels). In R17H02 (top center) two uniPNs with dendrites branching into DA1 (arrows) and one neuron branching into VA6 (arrowhead) were labeled. No GFP+ PN branched in VA1lm. In R28H10 (bottom center) a single uniPN with dendrites branching in VC1 was GFP+ (arrow). (b) Tracing the neuronal connections from the ORNs expressing the ligand CD19::sdc in identified glomeruli by using the R17H02 (top panels) and R28H10 (bottom panels) LexA drivers. Left: CD19::sdc+ axons from ORNs (red) driven by R17H02- and R28H10-LexA branch in the identified glomeruli (stippled circles). In the VA6 glomerulus of R17H02 and VC1 of R28H10, the CD19::sdc expression level was too low to be detected by immunostaining, but the expression in DA1 and VA1lm of R17H02 was above the detection level. Center: Induction of CD4::tdGFP expression in PNs triggered by CD19::sdc+ ORNs (red, left panels). In R17H02 (top center) there were uniPNs projecting to DA1 (arrows) and VA6 (arrowhead). In R28H10 (bottom center) a single uniPN projecting to VC1 was GFP+ (arrow). Expression of CD19::sdc with R28H10 and R17H02 induced GFP expression in a single multiPN (yellow arrowhead). (c) The diagrams show the induction pattern of uniPNs from (a) and (b). The multiPNs detected in (b) are not included. Scale bar = 50 µm.

DOI: https://doi.org/10.7554/eLife.32027.012

**Table 1.** Summary of the induction results in the identified glomeruli and circadian neurons.

(a and b) Summary of PNs labeled from ORNs expressing nSyb::CD19 (a) or CD19::Sdc (b) in identified glomeruli using the R17H02 and R28H10 LexA drivers. *# of GFP+ PNs*: number of PNs with GFP induction (median ±sd). *% of positive ALs*: percentage of antennal lobes with GFP+ PNs in identified glomeruli. *# of PNs reported*: numbers of PNs innervating identified glomeruli as reported in previous works (*Grabe et al., 2016*). In the column of R17H02, only the results from females (F) are listed because expression of the R17H02-lexA driver was highly variable in males. *n*: number of antennal lobes analyzed. *: one *R17H02>nSyb::CD19* and two *R17H02>CD19::Sdc* VA6 glomeruli were GFP+ (there were GFP+ dendrites branching in VA6), but were excluded because the cell bodies of the uniPN projecting into VA6 could not be identified. *$*: with nSyb::CD19 there is GFP induction in multiPNs in animals homozygous for the ligand (hom), but not in heterozygotes (het). (c) Summary of neurons with GFP expression in animals with donor PDF neurons expressing nSyb::CD19. *# of GFP+ Ns*: number of the different types of neurons labeled with GFP (mean ±s.e.m). *# of Ns reported with connections*: previous works did not report these data. *% of positive hemispheres*: percentage of hemispheres analyzed that contained GFP+ neurons. *n*: numbers of the hemispheres analyzed.

| A | GH146-nlgSNTG4 + LexAop-nSyb::CD19 | | | | |
|---|---|---|---|---|---|
| | PNs | # of GFP+ PNs | # of PNs reported | % of positive ALs | n |
| R17H02 | uniPN->DAl | 10.5±1.65 | 8±1.9(F) | 100 | 10 |
| | uniPN->VA6 | 1±0.33 | 1±0.4 (F) | 100 | 9* |
| | uniPN->VA1lUm | 0 | 6±0.9 (F) | 0 | 10 |
| | multiPN | 0 (het)$ 1±0.00 (horn) | 1 | 0 100 | 10 6 |
| R28H10 | uniPN->VCl | 1±0.00 | 1-2 | 100 | 22 |
| | multiPN | 0 (het)$ 1±0.48(hom) | 1 | 0 70 | 22 10 |
| B | GH146-nlgSNTG4 + LexAop-CD19::Sdc | | | | |
| | PNs | # of GFP+ PNs | # of PNs reported | % of positive ALs | n |
| R17H02 | uniPN->DAl | 11.5±11.43 | 8±+1.9 (F) | 100 | 10 |
| | uniPN->VA6 | 1±0.35 | 1±10.4 (F) | 100 | 8* |
| | uniPN->VA1lm | 0 | 6±0.9 (F) | 0 | 10 |
| | multiPN | 1±0.52 | 1 | 60 | 10 |
| R28H10 | uniPN->VCl | 1±0.31 | 1-2 | 90 | 20 |
| | multiPN | 1±0.39 | 1 | 90 | 20 |
| C | nSybE-nlgSNTG4 + LexAop-nSyb::CD19 | | | | |
| | types of neurons | #of GFP+Ns | # of Ns reported with connections | % of positive hemispheres | n |
| pdf | DN2 | 1±0.75 | N.A. | 55.6 | 18 |
| | DN3 | 7±2.3 | N.A. | 100 | 18 |
| | PER- | 7±2.81 | N.A. | 100 | 18 |

DOI: https://doi.org/10.7554/eLife.32027.013

(*Cavanaugh et al., 2014*; *Seluzicki et al., 2014*) and DN2 neurons (*Tang et al., 2017*) in the adult fly. In addition, it has been reported that s-LNv neurons make weak connections with LNd neurons that vary throughout the day (*Gorostiza et al., 2014*). We tested whether we could use TRACT, first, to confirm the connection between s-LNv and DN1 and/or LNd neurons, and second, to identify new synaptic partners of s-LNvs in other brain areas. We used a *pdf*-lexA driver to express the nSyb::CD19 or CD19::sdc ligands, performed immunostaining against the OLLAS tag and confirmed that the ligands were selectively expressed in the axon terminals of the PDF neurons (*Figures 5b* and *6* and *Figure 5—figure supplement 1*).

In the flies carrying *pdf*>nSyb::CD19, nSybE-nlgSNTG4 and UAS-CD4::tdGFP, we observed several neurons located in the central brain that expressed GFP (*Figures 5b* and *6* and *Table 1c*). Consistent with a previous work (*Tang et al., 2017*) indicating the s-LNvs make synaptic contacts with DN2 neurons, we performed immunocytochemistry using an antibody against the PER protein and observed GFP+, PER+ neurons in the DN2 cluster in 10 out of 18 hemispheres (*Figures 5b* and *6b*, and *Table 1c*). There are two PER neurons in the DN2 cluster (*Helfrich-Förster et al., 2007*). In

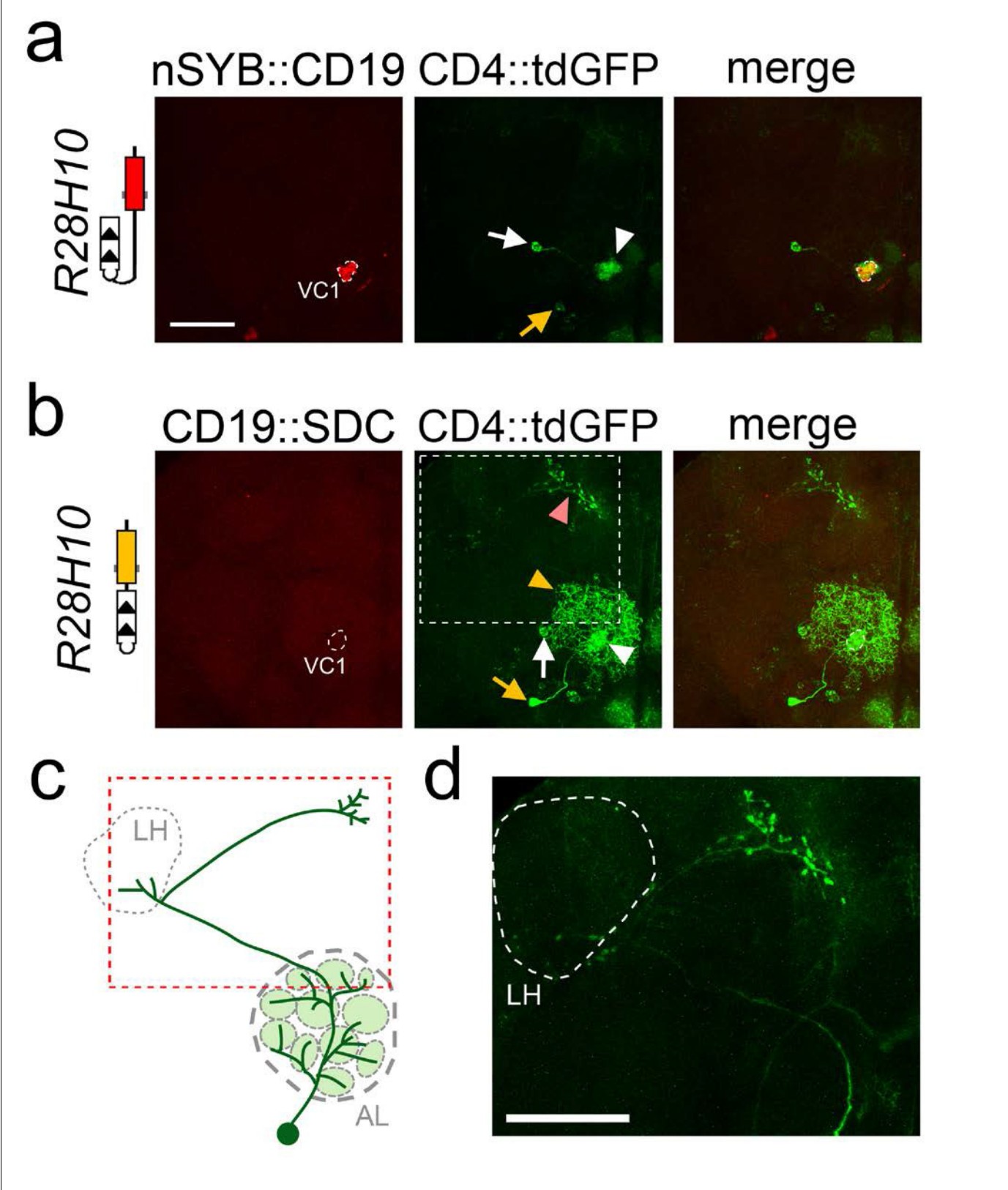

**Figure 4.** Labeling of one uniPN and one multiPN connected to glomerulus VC1 with nSyb::CD19 and CD19::sdc. Expression of nSyb::CD19 (homozygote, top panels) and CD19::sdc (heterozygote, middle panels) in VC1 glomerulus induced GFP expression in a single uniPN (white arrow) and one multiPN (yellow arrow). With the nSyb::CD19 ligand (**a**), both the cell body of the uniPN (white arrow) and its dendrite branching in the VC1

*Figure 4 continued*

glomerulus (white arrowhead) are visible. In contrast, the cell body of the multiPN is indicated by a yellow arrow. With the CD19::sdc ligand (**b**) the induction of GFP in the uniPN is comparable to that observed with nSyb::CD19 (white arrow and arrowhead point to the cell body and dendrite of uniPN, respectively). However, with CD19::sdc induction in the multiPN is stronger than with nSyb::CD19, and GFP labels the cell body (yellow arrow), its dendrites (yellow arrowhead), and its axon (pink arrowhead). High magnification image of the axon (white stippled inset) is shown in (**d**). (**c**) The diagram illustrates the position of the cell body of the multiPN with respect to the antennal lobe, and the trajectory of its axon (red stippled inset), projecting first into the LH (grey stippled circle), and then to the rostral part of the brain. This multiPN has been described previously - see (*Marin et al., 2002* panel H, *Figure 3*). All images are maximum projection confocal stacks. Scale bar = 50 µm.

DOI: https://doi.org/10.7554/eLife.32027.014

our experiments, in seven hemispheres we observed one GFP+, PER+ DN2 neuron, and in three hemispheres we observed two GFP+, PER+ DN2 neurons (*Figures 5b* and *6b*, and *Table 1c*). In contrast to previous works, we did not observe any GFP+ cells in the DN1 cluster but we observed several GFP+ cells that were PER-, close to the DN1 cluster (*Figures 5b* and *6a*) (*Cavanaugh et al., 2014*) (*Seluzicki et al., 2014*). In addition, we did not detect any induction in the LNd neurons (*Figure 5b*) (*Gorostiza et al., 2014*). Finally, we observed GFP induction in some of the DN3 neurons in all the brains we analyzed (*Table 1c*), and we confirmed that all the GFP+ cells in DN3s were also PER+ (*Figures 5b* and *6b*).

We observed that whereas some of the sets of neurons that were identified as potential synaptic partners with CD19::sdc were similar to those detected with nSyb:CD19, there were also some differences (*Figures 5b* and *6*, and *Figure 5*, *Figure 5—figure supplement 1*). Like nSyb::CD19, CD19::sdc also revealed potential synaptic connections with neurons in the DN2 and DN3 cluster (*Figure 5*, *Figure 5—figure supplement 1*). However, the numbers of neurons labeled with CD19::sdc were fewer than the ones with nSyb::CD19. This might be due to the fact that CD19::sdc driven by the pdf driver was expressed at very high levels in the cell body, but at much lower levels in the presynaptic terminals (*Figure 5*, *Figure 5—figure supplement 1*). In contrast, nSyb::CD19 was clearly detectable in the presynaptic terminals (*Figure 5b*). In addition, nSyb::CD19, but not CD19::sdc, detected some PER- neurons near the DN1 cluster, and, with CD19::sdc, but not with nSyb::CD19, we observed GFP induction in the l-LNvs and s-LNvs (the pdf neurons themselves) (*Figures 5b* and *6a* and *Figure 5*, *Figure 5—figure supplement 1*). Finally, we observed that the patterns of GFP induction between animals in this circuit were more consistent with nSyb::CD19 than with CD19::sdc (*Figure 5*, *Figure 5—figure supplement 1*). These observations indicate that TRACT can be used to discover candidate synaptic partners in an unbiased manner. Whereas the overall pattern of transneuronal labeling observed is broadly comparable when using the sdc or nSyb ligands, in some cases there may be potential synaptic partners that may be specifically revealed by one of the ligands, and that further experiments will be required to validate these putative connections.

## Discussion

Our experiments demonstrate that it is possible to take advantage of the molecular mechanisms of ligand-induced intramembrane proteolysis to trace neuronal circuits. We have generated transgenic animals in which neurons expressing an artificial ligand ('donor' cells) activate a genetically modified receptor on their synaptic partners ('receiver' cells). Using this system, called TRACT (for *TR*ansneuronal *AC*tivation of *T*ranscription) we have shown that expressing the ligand in a set of donor neurons can activate transcription in synaptically connected neurons in the *Drosophila* brain, in an anterograde manner. Using TRACT, we have confirmed the connectivity between ORNs and PNs in the antennal lobe, and have discovered new potential connections between PDF and PER neurons in the circadian circuit.

There are several advantages of transneuronal tracing based on ligand-induced membrane proteolysis. (1) TRACT is fully genetically-encoded, it only requires three constructs (a ligand, a receptor, and a reporter) and can be used with high reproducibility in transgenic animals. Moreover, as we demonstrate here, the ligand and/or the receptor can be driven with promoters specific to selective neuronal populations to reveal circuits of synaptically connected neurons. (2) In principle, it can be used in any species amenable to transgenesis. This feature is particularly advantageous for

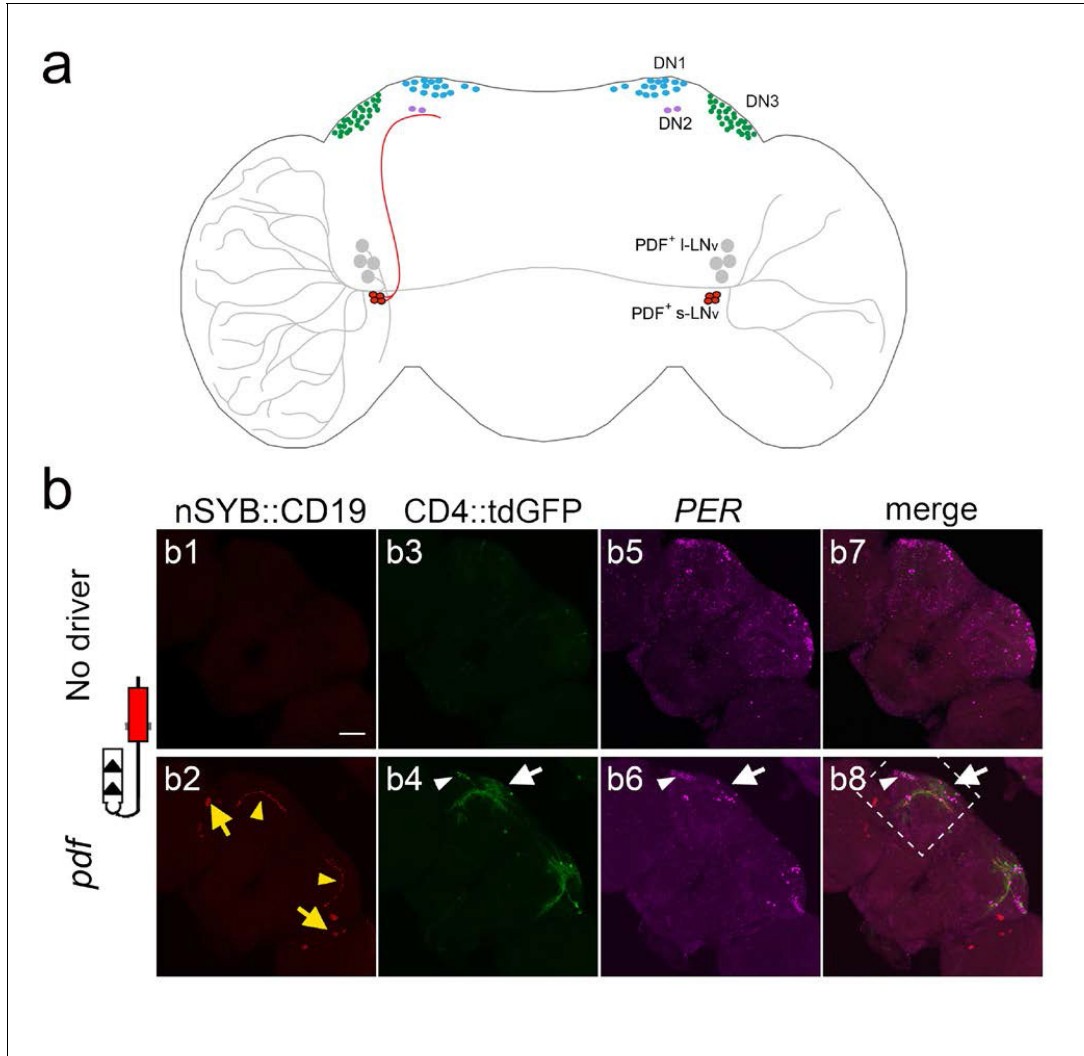

**Figure 5.** Using TRACT to identify synaptic targets of the PDF neurons in the central brain. (**a**) Diagram of *Drosophila* circadian neurons. The two groups of PDF LNvs are located close to optic lobe: four l-LNvs (black) have dendrites that branch into the ipsilateral optic lobe, and project their axons across the central brain to the contralateral optic lobe. Four s-LNvs (red) project their dorsal axons to dorsoposterior part of the central brain, where DN1 (blue), DN2 (purple) and DN3 (green) neurons are located. (**b**) Labeling of neurons that receive synaptic input from PDF cells expressing no ligand (top panels: b1, (**b3, b5, b7**), or the ligand nSyb::CD19 driven by the *pdf*-LexA driver (bottom panels: b2, (**b4, b6, b8**). To unbiasedly identify downstream synaptic targets of PDF neurons, we used the pan-neuronal receptor strain, nSybE-nlgSNTG4. b1, b2: nSyb::CD19 +expression pattern. b1: control brain with no nSyb::CD19 expression. b2: nSyb::CD19 driven by *pdf*-LexA accumulated (red) in the cell bodies (yellow arrows) and the axon terminals of s-LNv dorsal axons (yellow arrowheads). b3: No GFP induction was observed in the control brains without *pdf* >nSyb::CD19 ligand. b4: Induction of CD4::tdGFP expression in *pdf* >nSyb::CD19 brain in the vicinity of the trajectory of nSyb::CD19 +axons through the central brain. Arrow and arrowhead point to GFP +neurons in the DN1 region, and DN3 regions, respectively. b5, b6: Immunostaining with anti-PER antibody identifies DN1 (white arrow), DN2, and DN3 (white arrowhead) neurons. b7, b8: Merged images of PER (magenta), GFP (green) and nSyb::CD19 (red). Stippled square in b8 indicates region shown at high magnification (40X) in *Figure 6*.

DOI: https://doi.org/10.7554/eLife.32027.015

The following figure supplement is available for figure 5:

**Figure supplement 1.** Potential synaptic targets of the PDF neurons in the central brain using the CD19::Sdc ligand.

DOI: https://doi.org/10.7554/eLife.32027.016

*Drosophila* (*Bellen et al., 2010*; *St Johnston, 2002*), mice (*Anderson and Ingham, 2003*; *Kile and Hilton, 2005*) and zebrafish (*Fetcho and Liu, 1998*; *Patton and Zon, 2001*), three model organisms of great interest to neurobiologists, with an extensive arsenal of genetic tools. (3) The synaptically connected neurons can be studied in vivo with electrophysiological recordings, live imaging, and

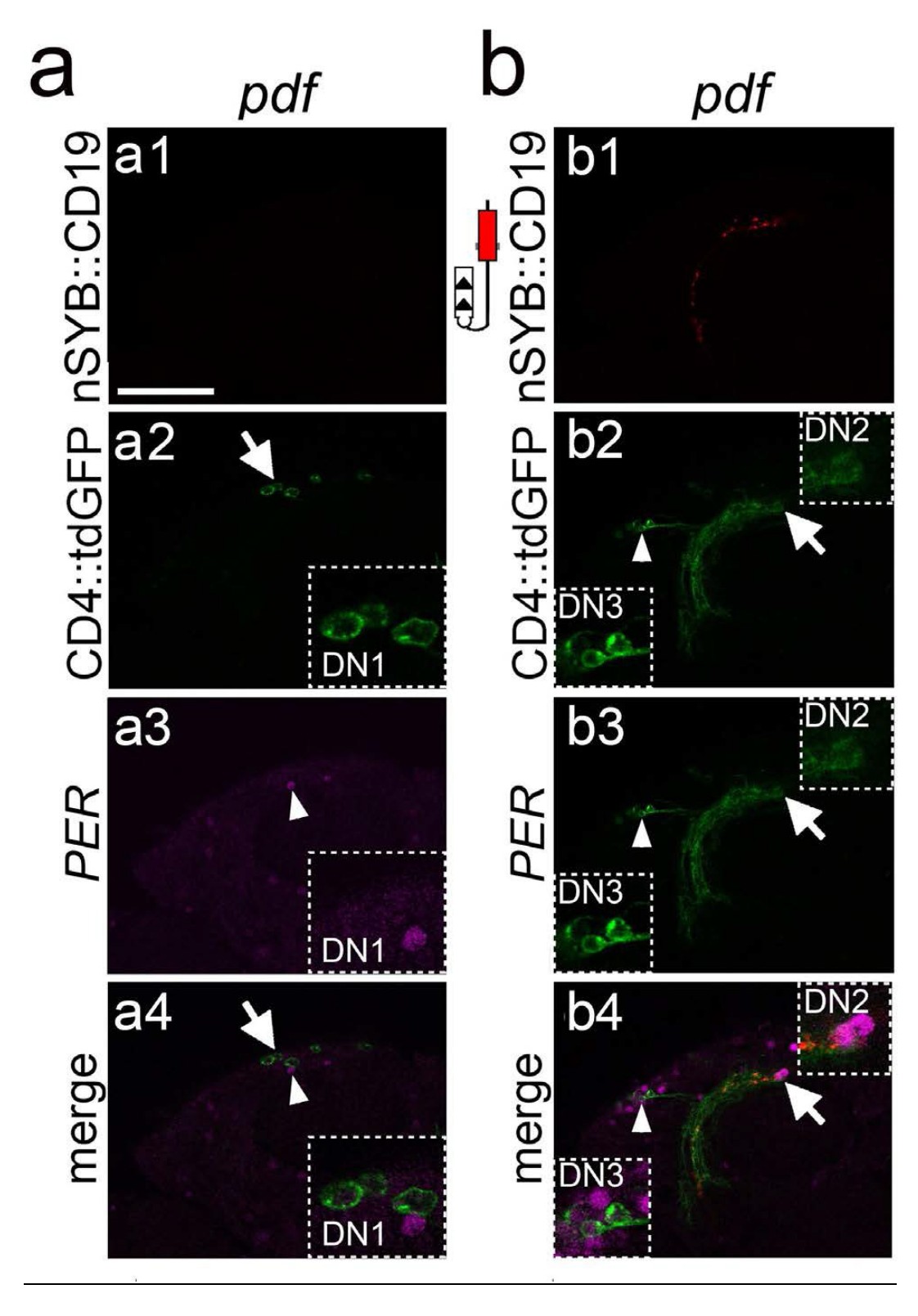

**Figure 6.** Characterization of the synaptic targets of the PDF neurons in the central brain. (**a and b**) Single optical confocal images at different depths of the brain showing a high magnification view of the stippled rectangle in *Figure 5b* bottom panels. (**a**) Induction of GFP in neurons around the DN1 region. a1: In this focal plane the nSyb::CD19 expression is not visible. a2: Induction of GFP expression in cells (arrow) around the DN1 cluster. Stippled square at the bottom right is a high magnification image of the GFP +cells. a3: PER immunostaining of the DN1 neurons. Arrowhead points to

*Figure 6 continued on next page*

*Figure 6 continued*

PER +cells shown at high magnification on the right bottom inset. a4: Merged image showing that the GFP +cells are not the PER +DN1 neurons. (b) Induction of GFP in cells in the DN2 and DN3 regions. b1: Axons from nSyb::CD19 (red). b2: Arrowhead and arrow point to cell bodies of GFP induced neurons in DN3 and DN2 regions, shown at higher magnification on the bottom left, and top right insets, respectively. b3: Immunostaining of PER (magenta). b4: Merged image showing that in the DN3 (bottom left inset) and DN2 (top right inset) regions, the GFP +neurons are also PER+. Scale bar = 50 µm.

DOI: https://doi.org/10.7554/eLife.32027.017

optical monitoring of activity, or in fixed tissue, combined with light or electron microscopy. (4) The system can be used in high-throughput experiments because, unlike electron microscopy (*Mikula and Denk, 2015*; *Bock et al., 2011*; *Tapia et al., 2012*), it is not labor intensive or time-consuming. This feature would be very useful to perform genetic screens seeking to identify mutations that affect neuronal connectivity, and to investigate how neuronal connections change in response to behavioral experience, diseases, or exposure to different environmental variables, such as drugs or toxic chemicals. (5) The system can be used to induce the expression of transgenes in the synaptically connected cells that allow for imaging of neuronal activity, such as genetically encoded $Ca^{2+}$ sensors (*Chen et al., 2013*) or for experimental modification of neuronal activity, such as optogenetic tools (*Kim et al., 2017*) or ion channels (*Lin et al., 2010*). (6) TRACT could be used to control neuronal function by regulating endogenous genes indirectly through nuclear translocation of molecules such as Cre, Flp, LexA, QF2, or TetA (*Riabinina et al., 2015*; *Sauer and Henderson, 1988*; *Dymecki, 1996*; *Lewandoski, 2001*; *Venken et al., 2011*; *del Valle Rodríguez et al., 2011*) or directly by fusing endogenous transcription factors to the artificial receptor. (7) In the experiments presented here, we localized the ligand in presynaptic sites and observed anterograde tracing, in a presynaptic to postsynaptic neuron direction. In principle, it should be possible to design a retrograde tracing system by localizing the ligand selectively in the postsynaptic sites (*Sheng and Kim, 2011*), and/or the receptor in the presynaptic site (*Südhof, 2012*). (8) Finally, we demonstrate that directing expression of the ligand into a subset of donor neurons localized to a restricted area of the nervous system activates transcription in a very selective subset of neurons that receive synaptic contact from those donor neurons. For example, we were able to activate transcription selectively in specific subsets of antennal lobe PNs that receive synaptic input from individual glomeruli. This observation indicates that even if there are no specific promoters capable of directly driving expression of transgenes into certain neuronal types (such as antennal lobe uniPNs that receive synaptic input from glomerulus VC1 and DN3 neurons), this strategy makes it possible to genetically manipulate highly specific populations of neurons based not on the genes that they express, but on the cells from which they receive synaptic input.

In any strategy for mapping synaptic connectivity there are two key problems that are important to be aware of. First, there may be neurons connected by synapses but the system fails to detect that they are connected (false negatives). For example, it is probably easier to miss the connection between neurons that have very few synapses between them. Second, there may be neurons that are not connected by synapses, but the system indicates that they are connected (false positives). For example, it is possible that some of the connections revealed by the tracing system could be between neurons that are close to each other, but not connected by synapses. This is a problematic scenario, as designing models of computations by brain circuits will be completely inadequate if they include synaptic connections that do not exist. This indicates that with any of the currently available methods to study neuronal connectivity, it is advisable to confirm that any connections revealed are validated by other complementary methods.

Our results of the connectivity between ORNs and the antennal lobe with TRACT are consistent with the published literature. First, with the *nSyb* and *sdc*-targeted ligand, we never observed any uniPN whose dendrites projected outside of the glomeruli where the ligand was expressed, thus indicating that TRACT has a very low rate of false positives. Second, with these ligands we consistently observed a number of uniPNs consistent with the published works, suggesting that for the connectivity between ORNs and uniPNs, TRACT has a very low rate of false negatives. However, we could only detect the expected multiPN (*Marin et al., 2002*) (*Figures 3b* and *4* and *Table 1*) in approximately 50% of the cases. Several reasons could account for the variable labeling of multiPNs that we observed in our data. First, we do not know whether the GH146 enhancer drives expression

of the nlgSNTG4 receptor into this multiPN in a consistent manner. If the expression of the receptor is variable between animals, this could explain why we detect the multiPN in some brains but not others. Second, whereas uniPNs have all their synapses concentrated in a single glomerulus, multiPNs have synapses distributed throughout multiple glomeruli. This means that the number of synapses between the multiPN and the ligands presented by ORN axons in an individual glomerulus is likely lower than between those of uniPNs and ORNs. This suggests that TRACT probably can detect contacts between ORNs and uniPNs with a higher sensitivity than it can between ORNs and multiPNs.

Our analysis of the connectivity of the PDF neurons in the circadian circuit using TRACT confirmed a recent report indicating a connection between PDF and DN2 neurons (*Tang et al., 2017*). In addition, using TRACT with the nSyb::CD19 and CD19::sdc ligand revealed that DN3 neurons are new potential postsynaptic targets for PDF neurons, an observation consistent with recent studies demonstrating that applying PDF causes a delay in the circadian phases of $Ca^{2+}$ activity in DN3 (*Liang et al., 2017*). The role of DN3s in the control of circadian rhythm is still poorly understood, and it would be interesting to use TRACT to selectively manipulate these labeled DN3s to investigate their function in regulating circadian behavior. It is important to note that although the axons from the PDF neurons (l-LNvs and s-LNvs) project very widely through the brain, we identified a handful of putative synaptic targets in brain regions that are consistent with a circadian function. These observations suggest that TRACT probably has a low rate of false positives, consistent with our data on the connectivity of the antennal lobe. Finally, our experiments suggest that TRACT may have some false negatives as it failed to reveal the connections between s-LNvs and DN1s, and between s-LNvs and LNds that have been postulated by previous works using GRASP (*Cavanaugh et al., 2014*; *Seluzicki et al., 2014*; *Gorostiza et al., 2014*). Several reasons could account for this discrepancy. First, in the previous published works, neither of the two components of the GRASP system were localized to synaptic sites, so it is possible that the GRASP signal that was detected could be due to proximity between axons of PDF and DN1 or LNd neurons that were not connected by synapses. Resolving this issue will require validating the connectivity between PDF and DN1 neurons using complementary methods, including electrophysiological methods or electron microscopy. Second, although the nSybE driver is supposed to be pan-neuronally homogeneous, it is likely that it may express transgenes in certain neuronal populations at weaker levels. For example, If the expression of the receptor is weak in DN1 and LNd neurons, this could explain why we did not detect any connections between s-LNvs and DN1 and LNd neurons. Third, it is possible that in our current implementation of the system, TRACT can reveal neurons connected by certain synapses but not others. PDF neurons are characterized by the expression of the neuropeptide PDF, and it is possible that the secretion of neurotransmitters and neuropeptides could occur at different presynaptic sites. It is possible that in this current implementation, TRACT may be able to detect synaptic contacts mediated by neurotransmitter vesicles, but perhaps it may fail to detect neuropeptide release sites. Further tests of the system in multiple circuits would be required to fully assess the sensitivity of TRACT.

Our results indicate that the choice of the domain used to target the ligand into the presynaptic sites determines the specific potential synaptic partners identified by TRACT. In initial experiments (described in the supplementary materials) we observed that fusing the CD19 ligand to the intracellular domains of several molecules known to be localized in presynaptic sites (including syntaxin, dpr-10, dip- ɣ, and neurexin) was sufficient to enrich the localization of the TRACT ligand into presynaptic compartments. However, when using the stringent test afforded by the anatomical specificity of uniPNs (whose dendrites branch into a single glomerulus where they receive synapses from ORNs expressing a single olfactory receptor molecule), we observed that using these ligands we could detect GFP expression in uniPNs whose dendrites branched in glomeruli other than the one where the TRACT ligand was expressed. These results suggest that the domains that we used from syntaxin, dpr-10, dip- ɣ, and neurexin were not useful to restrict transneuronal labeling between neurons that were exclusively connected by synapses (as opposed to any other non-synaptic forms of cell-cell contact, including mere proximity between their respective plasma membranes). In contrast, we observed that fusing the CD19 ligand to the intracellular domains from sdc and nSyb enables reliable transneuronal tracing that was restricted to neurons connected by synapses, as assessed by the stringent test of the branching of uniPNs' dendrites into single glomeruli. We did not observe any difference in the number or distribution of uniPNs labeled with nSyb::CD19 or

CD19::sdc in the antennal lobe. However, we observed that whereas a single copy of CD19::sdc was sufficient to label multiPNs, two copies of nSyb::CD19 were necessary to produce any multiPN labeling. This observation indicates that both ligands are effective at detecting both uni- and multiPNs. However, in some cases the amount of the ligand could be a limiting factor to the intensity of the labeling.

In the circadian circuit we observed that whereas the overall pattern of transneuronal labeling revealed by TRACT was similar with nSyb::CD19 and CD19::sdc, there were some differences.

First, we observed that the patterns of GFP induction were more consistent between animals with nSyb::CD19 than with CD19::sdc (*Figures 5b* and *6* and *Figure 5*, *Figure 5—figure supplement 5*). Second, we observed that some neuronal populations were labeled with one of the ligands, but not the other. Most notably, nSyb::CD19 (but not CD19::sdc) labeled some PER- neurons near the DN1 cluster and CD19::sdc (but not nSyb::CD19) labeled l-LNvs, and s-LNvs (the pdf neurons themselves). Several reasons could account for these differences, including the different biological functions of nSyb and sdc. nSyb is a molecule predominantly localized into the synaptic vesicles, which is displayed on the plasma membrane after synaptic vesicle fusion. Sdc is a heparan sulfate proteoglycan that in neurons is enriched in presynaptic sites. Thus, given the different biological functions of sdc and nSyb, and their specific subcellular localizations, it is expected that their domains may target expression of the TRACT ligands into slightly different locations in the presynaptic site, or they may influence its abundance on the synaptic cleft. Finally, it is possible that creating a hybrid molecule that combines, for example, some of the domains from a presynaptic marker and CD19 may perturb its targeting to the intended synaptic compartment, its abundance, or stability.

The two methods currently available to unambiguously confirm neuronal connectivity are dual single-cell electrophysiological recordings and serial electron microscopy. However, because single-cell recordings have a low throughput (in a typical experiment, only a handful of connections can be tested every day), this method is well suited to confirm the connections suggested by a different method, but it is not feasible to identify neuronal connections in an exploratory manner. Serial electron microscopy is an extremely powerful method to identify neuronal connections, but currently it has two main limitations. First, because it has a low throughput it cannot be used to study the connectivity of brain circuits across multiple animals. Second, the area that can be acquired in a single imaging event is restricted to a maximum of around 1 mm$^2$. This makes it well suited to investigate local connectivity, but extremely challenging to follow the connections between neurons located far away from each other, because they cannot be observed in a single imaging acquisition. The results shown here demonstrate that TRACT allows investigators to discover potential synaptic targets for genetically identified neurons in an unbiased manner, across multiple animals, thus allowing for high-throughput approaches. Then, the candidate synaptic partners identified by TRACT can be validated using complementary low-throughput methods that allow for unambiguous confirmation of connectivity, including dual single-cell electrophysiological recordings and/or electron microscopy.

While the present work was under review another work describing a genetic strategy to identify synaptically connected neurons called trans-TANGO was published (*Talay et al., 2017*). Trans-TANGO is the implementation of the TANGO system with a membrane-bound ligand, so that it can be applied to the study of neuronal connectivity. Both TRACT and trans-TANGO depend of ligand-induced proteolysis and subsequent release of a membrane anchored transcription factor. TRACT is based on intramembrane proteolysis by the endogenous ɣ-secretase, which is ubiquitously present in the plasma membrane of metazoan cells. Trans-TANGO depends on the cleavage and release of a transcription factor by the reconstitution of the viral protease TEV, which is triggered by the interaction between arrestin and the activation of an exogenous, engineered GPCR upon interaction with its ligand. The initial report of trans-TANGO also studied the connectivity between ORNs and antennal lobe neurons in *Drosophila* to validate its specificity. Our results in the *Drosophila* antennal lobe with TRACT, when the ligand was expressed into individual glomeruli, revealed connections between ORNs and uniPNs in the antennal lobe that are consistent with the published literature, but we only detected multiPNs in around 50% of the brains tested. In the trans-TANGO report with ligand expression into individual glomeruli, there was no evidence of detection of multiPNs, and the number of postsynaptic uniPNs observed with transTANGO was higher than those reported in previously published works. The authors concluded that the higher numbers of PNs observed with *trans*-Tango could represent false-positive signals that might have resulted from inefficient synaptic localization of the ligand due to its overexpression. Further experiments will clarify the respective rate of false

positives and false negatives by TRACT and transTANGO, and will allow for the optimization of these systems to enable the reliable application of these strategies for the investigation of neuronal connections in brain circuits.

In recent years, there has been a surge in interest for new methods to investigate synaptic connectivity in brain circuits (*Meinertzhagen and Lee, 2012*; *Denk et al., 2012*; *Bargmann and Marder, 2013*; *Swanson and Lichtman, 2016*). Identifying how neurons are connected is a valuable guide towards understanding how computations take place in the brain. In addition, recent research indicates that abnormal neuronal wiring might be the cause of several neurodevelopmental and psychiatric disorders, including autism and schizophrenia (*Peça and Feng, 2012*; *Rubinov and Bullmore, 2013*; *Mevel and Fransson, 2016*; *Narr and Leaver, 2015*). We anticipate that the advantages of TRACT will make it a useful addition to the arsenal of tools available to identify the wiring diagrams of brain circuits, and will open new avenues for enabling the genetic manipulation of neurons connected by synapses.

# Materials and methods

## Transgenic flies

*elav-nlgSNTG4:* The SNTG4 cassette is described in detail in our previous publication (*Huang et al., 2016*), and it contains a single chain antibody domain (S), the NRR domain from *Drosophila* Notch (N), the transmembrane domain from *Drosophila* Notch (T), and the esn variant of Gal4 (G4). nlgSNTG4 constructs were generated by ligating three PCR fragments of SNT, dNlg2 ICD and Gal4esn (esn, for short). The SNT and esn fragment was amplified by PCR from elav-SNTG4 (*Huang et al., 2016*), and the dNlg2 ICD was from *Drosophila* EST RH63339. These three fragments were subcloned into a P-element vector carrying the *elav* enhancer (*Yao and White, 1994*) using standard procedures. Transgenic elav-nlgSNTG4 flies were produced by standard P-element integration, were screened by GAL4 immunostaining, and the lines with the highest expression level of SNTG4 were chosen for subsequent experiments. Line #1, 2 and 4 were tested, and had similar results. Line #4 was used in this study.

GH146-nlgSNTG4: The nlgSNTG4 with V5 epitope was generated by amplifying nlgSNTG4 from elav-nlgG4 with a reverse primer that included the V5 epitope sequence, and inserted into pCasper-GH146QF (gift from C. Potter, Johns Hopkins University). Transgenic GH146-nlgSNTG4 flies were produced by standard P-element integration, were screened by V5 immunostaining, and the lines with the highest expression level of SNTG4 with PN specification, line #1, were chosen for subsequent experiments.

nSyb-nlgSNTG4: The nlgSNTG4 fragment with V5 epitope was directly amplified from GH146-nlgSNTG4, and was subcloned into pattNSYBBN (Addgene). Transgenic nSyb-nlgSNTG4 flies were produced by attb site-specific integration in the attP40 site.

LexAop-Syx::CD19: A fragment comprising the intracellular and transmembrane domains of syx was synthesized (Gene blocks, IDT inc.), fused to a fragment containing the extracellular domain of CD19 and the OLLAS epitope (Gene blocks, IDT inc.), and inserted into the LexAop pJFRC19 vector. Transgenic flies were produced by attb site-specific integration in attP2 site.

LexAop-nSyb::CD19: A fragment comprising the intracellular and transmembrane domains of nSyb was synthesized (Gene blocks, IDT inc), fused a fragment containing the extracellular domain of CD19 and the OLLAS epitope (Gene blocks, IDT inc.), and inserted into the LexAop pJFRC19 vector. Transgenic flies were produced by attb site-specific integration in attP2 site.

LexAop-CD19::Nrx1: A fragment comprising the CD19 extracellular domain followed by the OLLAS tag (Gene blocks, IDT inc.) was fused to a synthetic DNA fragment comprising the transmembrane and the intracellular domains of nrx1 (Gene blocks, IDT inc.) and inserted into pJFRC19. Transgenic flies were produced by attb site-specific integration in attP2 site.

LexAop-CD19::Dip: The extracellular domain of CD19 was fused to a synthetic full length open reading frame (ORF) of DIP γ followed by the OLLAS epitope, and inserted into the LexAop pJFRC19 vector. Transgenic flies were produced by attb site-specific integration in attP2 site.

LexAop-CD19::Dpr: The extracellular domain of CD19 was fused to the full length ORF of Dpr10 followed by the OLLAS epitope, and inserted into the LexAop pJFRC19 vector. Transgenic flies were produced by attb site-specific integration in attP2 site.

LexAop-CD19::Sdc: The extracellular domain of CD19 was fused to the full length ORF of DIP syndecan followed by the OLLAS epitope, and inserted into the LexAop pJFRC19 vector. Transgenic flies were produced by attb site-specific integration in attP2 site.

- 5xUAS-mCD8::GFP and 5xUASCD4::tdGFP reporters were gifts from Dr. Freeman, Oregon Health and Science University.

*pdf*-LexA (7M): gift from Dr. Rosbash, Brandeis University

orco-LexA::VP16: gift from Dr. Lee, Janelia Research Campus HHMI

Janelia LexA driver lines: GMR17H02, and GMR28H10-LexA were requested from Bloomington fly stocks.

## Genotypes of flies analyzed in the figures

*Figure 2e* top: 5XUAS-CD4::tdGFP, nSybE-nlgSNTG4/nSybE-nlgSNTG4; orco-LexA::VP16/LexAop-nSyb::CD19

*Figure 2e* bottom: 5XUAS-CD4::tdGFP/CyO; orco-LexA::VP16/LexAop-nSyb::CD19, GH146-nlgSNTG4#4

*Figure 3a* top: 5XUAS-CD4::tdGFP/GMR17H02-LexA; LexAop-nSyb::CD19/GH146-nlgSNTG4#4. We noticed that the male R17H02 has lower LexA activity in DA1 than the female. Therefore, only the female flies were analyzed.

*Figure 3a* bottom: 5XUAS-CD4::tdGFP/GMR28H10-LexA; LexAop-nSyb::CD19/GH146-nlgSNTG4#4

*Figure 3b* top: 5XUAS-CD4::tdGFP/GMR17H02-LexA; LexAop-CD19::Sdc/GH146-nlgSNTG4#4. Only the female flies were analyzed.

*Figure 3b* bottom: 5XUAS-CD4::tdGFP/GMR28H10-LexA; LexAop-CD19::Sdc/GH146-nlgSNTG4#4

*Figure 4a* 5XUAS-CD4::tdGFP/GMR28H10-LexA; LexAop-nSyb::CD19/GH146-nlgSNTG4#4, LexAop-nSyb::CD19

*Figure 4b and d* 5XUAS-CD4::tdGFP/GMR28H10-LexA; LexAop-CD19::Sdc/GH146-nlgSNTG4#4

*Figure 5b* top: nSyb-nlgSNTG4, 5XUAS-CD4::tdGFP/nSyb-nlgSNTG4; LexAop-nSyb::CD19/LexAop-nSyb::CD19

*Figure 5b* bottom and 8: nSyb-nlgSNTG4, 5XUAS-CD4::tdGFP/nSyb-nlgSNTG4, *pdf*-LexA; LexAop-nSyb::CD19/LexAop-nSyb::CD19.

All crosses were maintained in a 25C incubator with 12 hr-12hr dark-light cycles, and were repeated at least three times.

## Immunostaining and imaging of fly brains

After incubation at 29C for one day, the adult *Drosophila* were dissected, and the brains were removed in 1x PBS under a dissection microscope. For PER immunostaining, the flies were dissected one hour before the lights turned on (around 7 AM). Brains were fixed by immersing them in a 4% paraformaldehyde solution in PBS for 15 min at room temperature. Brains were washed in PBS three times for 10 mins each, followed by permeabilization with PBS + 0.5% triton X-100 (PBST) for 30 mins and blocking with 5% horse serum in PBST for 30 mins. The brain samples were stained with antibodies against GFP (rabbit polyclonals (AB3080 1:1000, AB3080P 1:1500 (Millipore)), or chicken polyclonal (Abcam), diluted at 1:1000, mcherry (rat monoclonal 5F8 (Chromotek) diluted at 1:1,000), Brp (mouse monoclonal nc82 (DSHB) diluted at 1:50), ChAT (mouse monoclonal 4B1 (DSHB) diluted at 1:200), GABA (rabbit polyclonal A2052 (Sigma) 1:200), PER (guinea pig polyclonal PA1140, gift from Dr. Sehgal, University of Pennsylvania), V5 (mouse monoclonal R960-25 (invitrogen) diluted at 1:300), OLLAS (rat monoclonal L2 NBP106713 (Novus) diluted at 1:300) diluted in 5% horse serum/PBST. Brains were Incubated with primary antibodies overnight at 4C, washed three times in PBST, incubated with secondary (goat secondary antibodies (Life Technologies) 1:500, except for rabbit anti-GFP AB3080P (Millipore), where the secondary was used at 1:750) for 1.5 hr at room temperature, washed in PBST and mounted on glass slides with a clearing solution (Slowfade Gold antifade reagent (Invitrogen)).

Stained brains were imaged with Confocal microscopes (Olympus Fluoview 300 or Zeiss 710) under 20x or 40X objectives. In a typical experiment, we imaged 150 sections with an optical

thickness of 0.3–0.5 µm from dorsal or ventral sides. Confocal stacks were processed with Fiji to obtain maximal projections.

## Generation of transgenic constructs

All constructs were generated using either PCR with NEB Phusion polymerase or IDT gene blocks. Cloning into transgenic vector plasmids was performed using Gibson assembly mix from NEB. The sequences of the constructs used to generate the data shown in main text (ID3 dNRR dNotch1 TMD dnlng2 esn V5, nSyb::CD19, and CD19::sdc) are shown in *Supplementary file 1*, including maps indicating their key elements).

## Acknowledgements

Supported by NIH grant UO1109147 from the BRAIN initiative.

## Additional information

### Funding

| Funder | Grant reference number | Author |
| --- | --- | --- |
| National Institutes of Health | U01 MH109147 | Carlos Lois |

The funders had no role in study design, data collection and interpretation, or the decision to submit the work for publication.

### Author contributions

Ting-hao Huang, Conceptualization, Data curation, Formal analysis, Validation, Investigation, Visualization, Methodology, Writing—original draft, Writing—review and editing; Peter Niesman, Formal analysis, Validation, Investigation, Visualization, Writing—review and editing; Deepshika Arasu, Donghyung Lee, Antuca Callejas, Validation, Investigation, Visualization; Aubrie L De La Cruz, Data curation, Validation, Investigation, Visualization, Writing—original draft, Writing—review and editing; Elizabeth J Hong, Experimental design, Analysis and data interpretation, Funding; Carlos Lois, Conceptualization, Resources, Data curation, Formal analysis, Supervision, Funding acquisition, Validation, Investigation, Visualization, Methodology, Writing—original draft, Project administration, Writing—review and editing

### Author ORCIDs

Ting-hao Huang (iD) http://orcid.org/0000-0002-2546-3525
Aubrie L De La Cruz (iD) http://orcid.org/0000-0002-7351-0285
Carlos Lois (iD) http://orcid.org/0000-0002-7305-2317

### Decision letter and Author response

Decision letter https://doi.org/10.7554/eLife.32027.022
Author response https://doi.org/10.7554/eLife.32027.023

## Additional files

### Supplementary files

• Supplementary file 1. ID3 dNRR dNotch1 TMD dnlng2 esn V5.
DOI: https://doi.org/10.7554/eLife.32027.018

• Supplementary file 2. Supplementary text for *Figure 2*.
DOI: https://doi.org/10.7554/eLife.32027.019

• Transparent reporting form
DOI: https://doi.org/10.7554/eLife.32027.020

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
