## [Decision Letter]

Thank you for submitting your article "Tracing neuronal circuits in transgenic animals by transneuronal control of transcription(TRACT)" for consideration by *eLife*. Your article has been reviewed by three peer reviewers, one of whom, Mani Ramaswami (Reviewer #1), is a member of our Board of Reviewing Editors, and the evaluation has been overseen by K VijayRaghavan as the Senior Editor.

The reviewers have discussed the reviews with one another and the Reviewing Editor has drafted this decision to help you prepare a revised submission.

Summary:

This paper builds on a system previously published by the same group (Huang et al., 2016) to label cells based on cell-to-cell contacts in vivo in *Drosophila*. In that publication the basic design was tested for a reporter of cell-cell contact, using an artificial chimeric receptor comprising a single chain antibody domain fused to parts of the *Drosophila* Notch receptor, which could release the Gal4 transcriptional upon binding of the CD19 ligand supplied by a contacting cell. While that publication showed proof of concept, it also revealed that the system was imperfect in that glial expression led to turning on of the Gal4 reporter in a limited subset of neurons only – rather than most, if not all, as one might actually expect.

The system is now re-engineered with the addition of synaptic targeting motifs to achieve anterograde-labeling of synaptic partners in the fly brain. This now results in more impressive reporting of projection neurons (PNs) as postsynaptic targets for selective olfactory receptor neurons (ORNs) expressing the ligand. For the most promising combination the authors do perform much needed quantification of efficiency and reproducibility (Table 1), which shows a degree of variability, though generally appears to perform to expectation.

This technique could be considered an incremental step from the author's previously published system, but the ability to label and manipulate neurons based on their connectivity in vivo in the brain has the potential to revolutionize the field. Also, while other systems to achieve the same goal are being / have been put forward (e.g. a very recent paper in Neuron from the Barnea lab), it will certainly take some time before the field works out the kinks, advantages and disadvantages of each strategy. Finally, because multiple systems doing the same thing will open up the potential of orthogonal approaches, this "Tools" manuscript is appropriate for publication in *eLife* after appropriate revisions, see below.

Essential revisions:

1) The paper contains an unusual amount of negative results and start-and-stop experimental lines that make it unnecessarily difficult to read. The text and data should be considerably condensed and a lot of the negative results could be briefly mentioned rather than shown. This is a tool-building exercise, and there is very little value of showing data using approaches or constructs that did not perform as expected and that will be abandoned; a brief mention of these failed approaches could even be confined to a table. The paper could be condensed into a much more readable 3-4 figure paper, showing data and controls for the 1-2 approaches that did produce the expected trans-synaptic labels and that the authors can conceivably recommend for use to the broader community.

As a guideline for the authors to consider, it seems that the first 3-4 main figures are not particularly compelling. Figure 1 is a drawing already published elsewhere in the same form, and only describes the basic concept; Figure 2 only contains a brain schematic and Figure 3 (and to some extent 4) show failed approaches. The authors produce and use a number of different pre-and post-synaptic constructs and direct their expression to a number of different cell types of the olfactory system. Instead of the generic schematic in Figure 1, the authors could perhaps produce a simplified diagram that accompanies each data figure and emphasizes the intra-cellular domain used for each experiment and the expression pattern of the pre- and post-synaptic partner. A simple color-code could show the expression in pre- and post- cell types, and different ball and stick models could identify the intracellular domains used. This would make it easier to understand what is going on in each figure.

2) The authors should include a figure with a proper characterisation of sub-cellular localisation of these reporters, e.g. when expressed in single cells together with a synaptic reporter, such as Brp-short-XFP for presynaptic sites.

3) As used, the olfactory system is attractive but its largely segregated nature might be a limited testbed. It is not clear how the authors can be certain that the postsynaptic cells they label via induced Gal4 activity are indeed postsynaptic to the ORNs expressing the receptors. For example, by using the GH146 promoter to limit expression of the synthetic receptor largely to PNs, aren't the authors biasing the outcome tremendously towards getting primarily PNs that are strongly connected or mainly confined to the particular glomerulus? The authors should consider adding in a figure addressing whether local neurons are revealed with this method if used in combination with the nSybE-nlgSNTG4? Or is trans-cellular Gal4 activation limited to extremely strongly connected cells? Is Gal4 activity seen in unexpected sets of cells.

4) The stainings seem of quite variable quality. What is the difference between Figure 6 and Figure 5 lower panel? How come VC1 is visible in one but not the other? In fact, the red labelling is inconsistent throughout. Glomeruli labelled by the same Gal4 line are expected to have similar amount of signal.

5) Related to this, Figure 5 shows no activation of PNs innervating VA1lm. The authors argue that this is because of low expression levels of the ligand. From the staining in Figure 5 it doesn't look like the ligand expression in VA1lm is lower than in VA6 and VC1. In fact, panel b seems to show more red signal in VA1lm than in VA6 or VC1. It would be important to know whether there is such a thing as a connection (in this case a glomerulus) that is refractory to transfer. Is it possible to achieve activation in VA1lm PNs using a different driver?

6) The authors claim that they have found a novel connection in the circadian circuit. However, in the absence of functional data this statement is overly strong and should be toned down.

7) Moreover, why are so few putative postsynaptic partners to the PDF-positive neurons. Published work from EM reconstruction efforts show an incredible degree of inter-connectedness that is simply not revealed here. If it turned out that this method biased toward strongly connected cells, that could be an advantage. But it should be discussed in detail if not experimentally characterised.

8) Figure 4 is called out inconsistently. The authors say that they observe projections to the MB and the LH with the nSybE-nlgSNTG4 and refer to Figure 4, but Figure 4 shows GH146-nlgSNTG4.

9) Can the authors elaborate on their choice of ligand for the experiments presented in Figure 6? The text seems to suggest that CD19::Sdc would be a better choice, but they use the nSyb::CD19 for mapping circadian neurons. It would be helpful to know if the two have been compared in this system.

10) The discussion should be expanded to compare and contrast this system, with the trans-Tango system that has been described in a paper from Gilad Barnea's group in Neuron published while this manuscript was in review.

---

## [Author Response]

Essential revisions:1) The paper contains an unusual amount of negative results and start-and-stop experimental lines that make it unnecessarily difficult to read. The text and data should be considerably condensed and a lot of the negative results could be briefly mentioned rather than shown. This is a tool-building exercise, and there is very little value of showing data using approaches or constructs that did not perform as expected and that will be abandoned; a brief mention of these failed approaches could even be confined to a table. The paper could be condensed into a much more readable 3-4 figure paper, showing data and controls for the 1-2 approaches that did produce the expected trans-synaptic labels and that the authors can conceivably recommend for use to the broader community.

We agree with these comments and have moved all the results with false positives into the supplementary material.

As a guideline for the authors to consider, it seems that the first 3-4 main figures are not particularly compelling. Figure 1 is a drawing already published elsewhere in the same form, and only describes the basic concept; Figure 2 only contains a brain schematic and Figure 3 (and to some extent 4) show failed approaches. The authors produce and use a number of different pre-and post-synaptic constructs and direct their expression to a number of different cell types of the olfactory system. Instead of the generic schematic in Figure 1, the authors could perhaps produce a simplified diagram that accompanies each data figure and emphasizes the intra-cellular domain used for each experiment and the expression pattern of the pre- and post-synaptic partner. A simple color-code could show the expression in pre- and post- cell types, and different ball and stick models could identify the intracellular domains used. This would make it easier to understand what is going on in each figure.

We have modified Figure 1 to include schematics of the different domains that are used throughout the text. As suggested, we have added the color-coded cartoons of the different ligands (nSyb and sdc) to facilitate the interpretation of the figures.

2) The authors should include a figure with a proper characterisation of sub-cellular localisation of these reporters, e.g. when expressed in single cells together with a synaptic reporter, such as Brp-short-XFP for presynaptic sites.

We have performed this experiment, and now we include a new supplementary figure where we demonstrate the following:

a) The ligand fused to neurexin leads to the distribution of the ligand somewhat enriched in the presynaptic sites, but also a clearly detectable amount of ligand along the distal part of the axons, outside of the presynaptic zone. This lack of strict synaptic localization is consistent with our observation that the ligand fused to neurexin labels neurons that are not connected by synapses.

b) In contrast, we demonstrate that when the CD19 ligand is fused to the intracellular domains of synaptobrevin (syb) or syndecan (sdc), the ligand is strictly localized in the neuropil region where BRP is present, and there is no ligand detectable along the sectors of the axons located outside of the glomeruli. Again, this result is consistent with our observation that the syb:CD19 and CD19:sdc ligands produce labeling of neurons that are strictly connected by synapses in the antennal lobe, as indicated by previous works.

3) As used, the olfactory system is attractive but its largely segregated nature might be a limited testbed. It is not clear how the authors can be certain that the postsynaptic cells they label via induced Gal4 activity are indeed postsynaptic to the ORNs expressing the receptors. For example, by using the GH146 promoter to limit expression of the synthetic receptor largely to PNs, aren't the authors biasing the outcome tremendously towards getting primarily PNs that are strongly connected or mainly confined to the particular glomerulus? The authors should consider adding in a figure addressing whether local neurons are revealed with this method if used in combination with the n-SybE-nlgSNTG4? Or is trans-cellular Gal4 activation limited to extremely strongly connected cells? Is Gal4 activity seen in unexpected sets of cells.

We have data demonstrating that the LNs are also induced with TRACT upon expression of the ligand in ORNs, and we have added a new supplementary figure showing these results. In these experiments we express the receptor into all neurons (using the nSybE driver), and the ligand with most ORNs (with the orco driver), and we observe GFP induction both in PNs and LNs. However, when LNs are labeled is it not possible to judge whether the system is truly synaptic specific. As we explain in the text, the critical test to assess synaptic specificity is whether uniPNs branch their dendrites exclusively into the individual glomeruli where the ligand is expressed. With uniPNs the test is unambiguous: if there are dendrites from PNs that branch outside of the ligand+ glomeruli the system is not specific for synapses. In contrast with uniPNs, the vast majority of LNs branch their dendrites in most (if not all) glomeruli. Thus, even if the system is specific for synaptic connections, when expressing ligand into a single glomerulus, the LNs will have dendrites branching into most (or all) glomeruli, and this makes it impossible to test the specificity of the system.

In summary: TRACT can detect the connectivity of ORNS to LNs and PNs, but only the ORN to PN allows us to test the specificity of the labeling.

4) The stainings seem of quite variable quality. What is the difference between Figure 6 and Figure 5 lower panel? How come VC1 is visible in one but not the other? In fact, the red labelling is inconsistent throughout. Glomeruli labelled by the same Gal4 line are expected to have similar amount of signal.

In the original Figure 5, we only used one copy of the nSyb::CD19 transgene while there were two copies of it in the original Figure 6, which increased the level of the ligand expression. We noticed that the OLLAS antibody we used to detect the ligand expression is not particularly sensitive. Therefore, certain low levels of the ligand expression cannot be visualized by immunostaining, even though they were still high enough to induce the GFP expression in the postsynaptic neurons.

5) Related to this, Figure 5 shows no activation of PNs innervating VA1lm. The authors argue that this is because of low expression levels of the ligand. From the staining in Figure 5 it doesn't look like the ligand expression in VA1lm is lower than in VA6 and VC1. In fact, panel b seems to show more red signal in VA1lm than in VA6 or VC1. It would be important to know whether there is such a thing as a connection (in this case a glomerulus) that is refractory to transfer. Is it possible to achieve activation in VA1lm PNs using a different driver?

We apologize, because we made a mistake in our original submission – we wrote “ligand” when we meant “receptor”. It is correct that VA1lm expressed ligand more strongly than in VA6 or VC1. However, the GH146 driver does not lead to expression of the receptor into all PNs, and it is possible that the PNs whose dendrites branch into VA1lm do not express the receptor, and thus, cannot be induced when the ligand is expressed into this glomerulus. We have corrected this mistake and explain that the failure to detect uniPNs innervating VA1lm could be due to a lack of expression of the receptor in the VA1lm uniPNs (in the original submission we stated that this could be due to a lack of expression of the ligand in VA1lm).

6) The authors claim that they have found a novel connection in the circadian circuit. However, in the absence of functional data this statement is overly strong and should be toned down.

We agree that this result needs to be confirmed, and we tone down the statement as follows:

Abstract: “..we have discovered potential new connections between neurons in the circadian circuit.”

Introduction: “..we have discovered that PDF neurons are connected to a potential new synaptic target,..”.

Results: (i) “…These observations suggest the potential existence of previously unknown connections between neurons….”. (ii)”. Future experiments will be needed to validate these putative connections”.

Discussion: “…TRACT revealed that DN3 neurons are new potential postsynaptic targets for PDF neurons..”

In addition, we have included a new paragraph in the Discussion section (see following paragraph) where we explain that it would always be advisable to validate the results obtained with TRACT with a complementary method, such as electrophysiological recordings or EM.

“The two methods currently available to unambiguously confirm neuronal connectivity are dual single-cell electrophysiological recordings and serial electron microscopy. […] Then, the candidate synaptic partners identified by TRACT can be validated using complementary low-throughput methods that allow for unambiguous confirmation of connectivity, including dual single-cell electrophysiological recordings and/or electron microscopy.”

7) Moreover, why are so few putative postsynaptic partners to the PDF-positive neurons. Published work from EM reconstruction efforts show an incredible degree of inter-connectedness that is simply not revealed here. If it turned out that this method biased toward strongly connected cells, that could be an advantage. But it should be discussed in detail if not experimentally characterised.

We agree with this concern, and have added the following comment on the discussion:

“Finally, our experiments suggest that TRACT may have some false negatives as it failed to reveal the connections between s-LNvs and DN1s, and between s-LNvs and LNds that have been postulated by previous works using GRASP(Cavanaugh et al., 2014) (Seluzicki et al., 2014) (Gorostiza et al., 2014). […] This observation indicates that both ligands are effective at detecting both uni- and multiPNs. However, in some cases the amount of the ligand could be a limiting factor to the intensity of the labeling.”

8) Figure 4 is called out inconsistently. The authors say that they observe projections to the MB and the LH with the nSybE-nlgSNTG4 and refer to Figure 4, but Figure 4 shows GH146-nlgSNTG4.

We apologize for this mistake and have now corrected it.

9) Can the authors elaborate on their choice of ligand for the experiments presented in Figure 6? The text seems to suggest that CD19::Sdc would be a better choice, but they use the nSyb::CD19 for mapping circadian neurons. It would be helpful to know if the two have been compared in this system.

As suggested by the reviewers, we have recently compared the nSyb and sdc ligands in the circadian system. When the nSyb and sdc ligands are expressed in pdf neurons we have observed that the pattern of GFP+ induced neurons overlap for most of the potential synaptic targets, but differ in others. We have included a new supplementary figure that shows the comparison between sdc and nSyb ligands in the circadian system, and have added a comment in the Results section describing these differences, and another comment in the Discussion section explaining potential reasons that could account for these differences.

“In the circadian circuit we observed that whereas the overall pattern of transneuronal labeling revealed by TRACT was similar with nSyb::CD19 and CD19:sdc, there were some differences. […] Finally, it is possible that creating a hybrid molecule that combines, for example, some of the domains from a presynaptic marker and CD19 may perturb its targeting to the intended synaptic compartment, its abundance, or stability.”

10) The discussion should be expanded to compare and contrast this system, with the trans-Tango system that has been described in a paper from Gilad Barnea's group in Neuron published while this manuscript was in review.

This is a difficult request, because we do not want to appear petty by criticizing their work. We have attempted to be as impartial as possible, and we mention two comparisons:

a) Whereas TRACT reveals antennal lobe multiPNs in 50% of the brain, there is no evidence for multi-PNs labeled with trans-TANGO.

b) TRACT appears to label the expected number of PNs, consistent with the literature. However, trans-TANGO appears to label more PNs than the expected number. To address this issue, we have copied verbatim the wording they used in the discussion of their results when trying to explain the discrepancy between the numbers of neurons that they observed and the numbers published in the literature [“….the higher numbers of PNs observed with trans-Tango could represent false-positive signals that might have resulted from inefficient synaptic localization of the ligand due to its overexpression. “].

We have included the following paragraph towards the end of the discussion to compare TRACT and trans- TANGO. We are happy if the reviewers and/or editors have a better suggestion about how to handle this issue.

“While the present work was under review another work describing a genetic strategy to identify synaptically connected neurons called trans-TANGO was published (Talay et al., 2017). […]Further experiments will clarify the respective rate of false positives and false negatives by TRACT and transTANGO, and will allow for the optimization of these systems to enable the reliable application of these strategies for the investigation of neuronal connections in brain circuits.”